# Multi-Detection Size Exclusion Chromatography as an Advanced Tool for Monitoring Enzyme–Antibody Conjugation Reaction and Quality Control of a Final Product

**DOI:** 10.3390/molecules28114567

**Published:** 2023-06-05

**Authors:** Adela Štimac, Tihana Kurtović, Beata Halassy

**Affiliations:** 1Centre for Research and Knowledge Transfer in Biotechnology, University of Zagreb, Rockefellerova 10, 10000 Zagreb, Croatia; 2Center of Excellence for Virus Immunology and Vaccines, Rockefellerova 10, 10000 Zagreb, Croatia

**Keywords:** size exclusion chromatography, advanced molecular characterization, multi-detection, IgG-HRP conjugate, ELISA, process control, quality control

## Abstract

The multi-detection size exclusion chromatography (SEC) has been recognized as an advanced analytical technique for the characterization of macromolecules and process control, as well as the manufacturing and formulation of biotechnology products. It reveals reproducible molecular characterization data, such as molecular weight and its distribution, and the size, shape, and composition of the sample peaks. The aim of this work was to investigate the potential and suitability of the multi-detection SEC as a tool for surveillance over the molecular processes during the conjugation reaction between the antibody (IgG) and horseradish peroxidase (HRP), and demonstrate the plausibility of its application in the quality control of the final product, the IgG-HRP conjugate. Guinea pig anti-Vero IgG-HRP conjugate was prepared using a modified periodate oxidation method, based on periodate oxidation of the carbohydrate side chains of HRP, followed by the formation of Schiff bases between the activated HRP and amino groups of the IgG. The quantitative molecular characterization data of the starting samples, intermediates, and final product were obtained by multi-detection SEC. Titration of the prepared conjugate was performed by the ELISA and its optimal working dilution was determined. This methodology proved to be a promising and powerful technology for the IgG-HRP conjugate process control and development, as well as for the quality control of the final product, as verified by the analysis of several commercially available reagents.

## 1. Introduction

Horseradish peroxidase (HRP, EC 1.11.1.7), found in horseradish roots, is the most widely used enzyme for the labelling of antibodies in many different immunoassays, including ELISA, Western blotting, and immunohistochemistry. It is a stable, inexpensive, and easily accessible ~44 kDa glycoprotein which has been used to detect biological compounds by measuring absorption, fluorescence, or luminescence [1,2,3]. The polysaccharide residues of the HRP molecule may be oxidized with sodium periodate to generate reactive aldehyde groups that can react with the amino groups of an antibody (IgG) through the formation of Schiff base intermediates. These relatively labile intermediates can be stabilized by reductive amination with sodium cyanoborohydride or sodium borohydride [3,4,5,6].

In antibody conjugation reactions, it is important to retain the antibody–enzyme complex activity to the greatest extent and prevent the formation of aggregates which reduce the amount of the active ingredient and lower the quality of the final product [1,7]. Therefore, an in-depth characterization of the protein sample and the detection/quantification of its aggregate share are highly recommended. Multi-detection size exclusion chromatography (SEC) has been recognized as a valuable analytical technique for the characterization of macromolecules and process control, as well as the manufacturing and formulation of biotechnology products. It provides a comprehensive insight into the molecular characterization data of the protein sample, including absolute molecular weight (*Mw*), molecular weight distribution or polydispersity, molecular size, intrinsic viscosity (*IV*), and recovery. Moreover, it is possible to obtain additional information on the macromolecular structure, conformation, aggregation, branching, and copolymer/conjugate composition [7,8,9,10].

Four detectors most often associated with an advanced multi-detection SEC system are the differential refractive index (RI) detector, the static light scattering (SLS) detector, the differential viscometer detector, and the UV detector. Multi-detection SEC removes the need for column calibration, but it is necessary to calibrate the detector responses by analyzing the stable protein standard with a defined concentration, *Mw*, refractive index increment (d*n/*d*c*), and molar extinction coefficient (d*A/*d*c*) in order to obtain the required detector calibration constants (*K_RI_*, *K_UV_*, *K_LS_*, and *K_Visc_*). The responses of the RI and UV detectors are directly proportional to the sample’s concentration (Equations (1) and (2)). The exact concentration can be calculated from the RI detector with the knowledge of the sample’s d*n*/d*c* and from the UV detector with the knowledge of the sample’s d*A*/d*c* [7,9,10,11,12].
(1)$$RI\;output\;\left(mV\right)=K_{RI}\cdot dn/dc\cdot c\cdot injection\;volume\;$$
(2)$$UVoutput\;\left(mV\right)=K_{UV}\cdot dA/dc\cdot c\cdot injection\;volume$$

The SLS detector responds to a sample’s molecular weight, and, through a combination of the RI and SLS detectors, the absolute *Mw* of the sample’s components can be calculated independent of the retention volume or any calibration standards (Equation (3)), as opposed to a column calibration system that can only provide a relative measurement (obtained *Mw* values are relative to the used standards) [7,9,10,11,12].
(3)$$LS\;output\;\left(mV\right)=K_{LS}\cdot Mw\cdot {\left(dn/dc\right)}^{2\;}\cdot c\cdot injection\;volume$$

According to the size exclusion theory, the principle of SEC is the separation of molecules based on their hydrodynamic size (function of both mass and structure), not molecular weight. Therefore, the *Mw* values determined by column calibration SEC and those determined by multi-detection SEC could be different because column calibration SEC introduces into the analysis shape-dependent inaccuracies. In practice, if the calibration curves were constructed with a series of two differently shaped molecules, they would yield different slopes [7].

Four SLS instruments are nowadays available, which differ in the angle at which the intensity of the scattered light is measured: right-angle light scattering (RALS), low-angle light scattering (LALS), hybrid RALS/LALS, and multi-angle light scattering (MALS). Each of them is slightly different and each has advantages and disadvantages. The hybrid RALS/LALS detector offers the advantages of RALS and LALS without any of their disadvantages, making it excellent for measuring the molecular weight of any sample type. The SLS detector response factors are used to calculate the distribution of the weight-averaged (*Mw*), number-averaged *(Mn*), and z-averaged (*Mz*) molecular weights. Using these three values, it is possible to obtain an insight into the entire molecular weight distribution. The *Mw*/*Mn* value corresponds to the polydispersity (*Pd*) of the sample and is related to the size distribution. The *Mw*/*Mn* ratio of 1 indicates that there is a symmetrical and homogeneous size distribution [13]. Furthermore, the SLS measurements show the power in the detection of trace amounts of high-molecular-weight soluble aggregates, because their LS signals have a higher intensity compared to UV and RI signals [9].

The differential viscometer detector measures the changing viscosity of the solution as the sample eluates and provides information about the structure. The *IV* of sample can be directly calculated with the parameters obtained from the RI and viscometer detectors (Equation (4)). Triple detection includes the RI or UV detector, SLS, and viscometer acting together, with each detector providing complementary but different information. Therefore, the hydrodynamic radius (*Rh*) can be indirectly calculated from the data of the RI, SLS, and viscometer detectors, and represents the radius of the theoretical sphere occupied by a sample of the known calculated *Mw* and *IV* [9,10,11,12].
(4)$$Viscometer\;output\;\left(mV\right)=K_{Visc.}\cdot IV\cdot c\cdot injection\;volume$$

In this study, we investigated the potential and suitability of the multi-detection SEC as a tool for monitoring the conjugation reaction between the antibody (guinea pig anti-Vero IgG) and HRP using the periodate oxidation method and employed it for the advanced molecular characterization of the prepared IgG-HRP conjugate. Starting samples (IgG and HRP), intermediates, and final product, as well as three different commercially available IgG-HRP conjugates, were analyzed. The absolute molecular weight, polydispersity, and composition of the sample peaks were determined, as well as the aggregate content of each final product. In addition, the enzymatic/immunological functionality of the conjugate was examined by the ELISA.

## 2. Results

### 2.1. Preparation of IgG-HRP Conjugate with Monitoring of the Conjugation Reaction by Multi-Detection SEC

The purity of guinea pig anti-Vero IgG sample used for the research was confirmed by multi-detection SEC analysis (Appendix A and Appendix A). Protein A affinity chromatography gave a product with ~92% monomers, ~2% dimers, and ~6% aggregate content, and without any contaminants, which is consistent with our previous results [14]. The obtained absolute *Mw* values for the IgG monomer (~153 kDa) and dimer (~300 kDa) were in line with the expected ones. Moreover, the IgG monomer polydispersity value (*Mw*/*Mn*) of 1.002 confirmed the molecular homogeneity and symmetrical size distribution. The obtained *Rh* value for the IgG monomer (Table 1) was in good agreement with the reported values [7,9].

Before conjugation, it was a necessity to transfer IgGs into a high-pH matrix since alkaline conditions enhance the formation of Schiff bases [5,15]. Multi-detection SEC results (Table 1, Appendix A, and Appendix A) confirmed that IgG molecular parameters were not affected by the alkaline pH. Moreover, the aggregate formation was not detected.

The HRP activation was performed in the dark to prevent sodium periodate breakdown and only for a limited period of time to avoid the loss of enzymatic activity [4,5]. Self-coupling of the activated HRP was minimized through the removal of the periodate excess by ultrafiltration against 1 mM sodium acetate buffer, pH 4.4 [4,16]. Aliquots of the HRP solution before and after activation were examined by multi-detection SEC (Appendix A and Appendix A). According to the obtained results, the oxidation of polysaccharide residues of the HRP did not affect its *Mw*, nor the polydispersity or composition of the sample peaks. The HRP purity was about 94% and the obtained absolute *Mw* value was in accordance with the theoretical one (Table 1).

After the periodate oxidation of the polysaccharide residues, Schiff bases formed between the amino groups of the IgG and the active aldehyde groups of the HRP in an alkaline pH were further converted into stable alkylamine linkages by a reduction with sodium borohydride [4,17]. Finally, ultrafiltration of the resulting IgG-HRP conjugate was performed, giving the final product (UF IgG-HRP).

The representative multi-detection SEC chromatogram of the final product, UF IgG-HRP (Figure 1), shows a peak with a retention volume (*V_R_*) at 8.6 mL, corresponding to the IgG-HRP conjugate, but also a presence of a large fraction of soluble aggregates with the *V_R_* <8 mL. The absolute *Mw* value for the IgG-HRP conjugate was ~235 kDa (Table 1 and Table 2), indicating that, on average, two activated HRP molecules (~43 kDa) were bound to one molecule of IgG (~153 kDa).

The quantitative properties of different molecular forms present in the conjugation reaction before and after the reduction step, as well in the final UF IgG-HRP sample, provided information on changes triggered by particular process steps (Table 2). The peaks with the *V_R_* at 6.8 mL in all samples were related to the heterogeneous population of higher-order aggregates, as confirmed by the relatively high polydispersity (>1.2). The peaks with the *V_R_* at 8.6 mL corresponded to the IgG-HRP conjugate with the average stoichiometry of 1:2, while those with the *V_R_* at 10.9 mL remained unidentified, but could possibly stem from impurities detected in the HRP sample, according to the obtained absolute *Mw* value (Appendix A and Appendix A). The reduction step did not affect the *Mw* or composition of the sample peaks. However, an increase of the aggregate share (>60%) in the IgG-HRP sample after ultrafiltration occurred. In addition, the qualitative comparison of all tested samples (Figure 2) and quantitative properties of the IgG-HRP conjugate (Table 2) showed that neither free IgG nor free activated HRP was present in the final product.

### 2.2. Evaluation of IgG-HRP Conjugate Functionality

The functionality of the prepared anti-Vero IgG-HRP conjugate (so far termed the UF IgG-HRP) was examined by the ELISA for the quantification of the total Vero cell lysate proteins. In the preliminary experiment, an anti-Vero IgG concentration, used for coating the plate, and an anti-Vero IgG-HRP conjugate dilution were varied to determine the optimal working dilution (Appendix A). The prepared conjugate was fully functional in the ELISA, as demonstrated by the excellent regression lines of the Vero lysate standard curves (in the range of 1–30 ng/µL) for the most suitable dilutions of the anti-Vero IgG-HRP conjugate (Figure 3). The optimal concentration of the anti-Vero IgG for plate coating was 10 μg/mL and the most appropriate dilution of the anti-Vero IgG-HRP conjugate (4.4 mg/mL) was 1:2000.

### 2.3. Multi-Detection SEC Analysis of Commercially Available IgG-HRP Conjugates

In order to compare the absolute *Mw*, polydispersity, and composition of sample peaks between various reagents (Table 3), we lastly examined three different commercially available IgG-HRP conjugates from two manufacturers: anti-goat IgG-HRP produced in rabbit (IgG-HRP 1), anti-rabbit IgG-HRP produced in goat (IgG-HRP 2), and anti-guinea pig IgG-HRP produced in rabbit (IgG-HRP 3). Their representative multi-detection SEC chromatograms are presented in Appendix A Appendix A, while the overlay of RI data, including RI data obtained for the prepared in-house UF IgG-HRP conjugate, is shown in Figure 4. The qualitative comparison (Figure 4) and quantitative properties of commercially available IgG-HRP conjugates (Table 3) showed the prevailing peak at 9.4 mL, which could correspond to the free IgG with an absolute *Mw* value of ~150 kDa and a share ranging between 42 and 62%.

From the obtained *Mw* for the IgG and HRP, and peaks with *V_R_* at 8.7 mL and 8.2 mL, we could estimate that the peaks with *V_R_* at 8.7 mL and 8.2 mL could correspond to the IgG-HRP conjugates with the average stoichiometry of 1:2 and 1:4 or 1:5, respectively. The obtained *Mw*/*Mn* values indicated a high degree of molecular homogeneity and symmetrical size distribution of those peaks. The total share for both types of IgG-HRP conjugates (with the average stoichiometry of 1:2 and 1:4 or 1:5) in each of the three tested samples was below 19%. All samples also contained a population of higher-order aggregates with the *V_R_* < 6.7 mL (share from 16–21%). The peaks with the *V_R_* at 10.8 mL (share from 8–18%) remained unidentified. The obtained *Rh* values for IgG-HRP conjugates (Table 3) increased compared to the IgG monomer, as expected, in accordance with the increase in the *Mw* of the conjugates and the estimated average stoichiometry of their components.

## 3. Discussion

An ideal coupling reaction procedure would be the one generating a homogeneous antibody–enzyme conjugate composed of one molecule of enzyme linked with one molecule of antibody in which most of the immunological and enzymatic activity is preserved. However, even when performing the reaction under strictly controlled conditions, the formation of undesirable conjugate types with different stoichiometric ratios could not be avoided. Therefore, it is necessary to pay attention to the retention of antibody and enzyme activity and the prevention of extensive aggregate formation, which reduces the amount of the active ingredient in the sample and may cause precipitation [1,18]. In this study, we aimed to demonstrate the potential of the multi-detection SEC in the advanced molecular characterization of the coupling reaction between IgG and HRP in order to gain insight into the process yield and the conjugate quality with respect to the preservation of its immunological and enzymatic activity. The IgG-HRP conjugate was prepared by the modified periodate oxidation method, which was first described by Nakane and Kawaoi [6]. The protocols for the periodate oxidation method described in the literature vary according to the concentration of the IgG, HRP, and required reagents, the NaIO_4_/HRP and HRP/IgG molar ratio, the type of reducing agent, the duration of the reaction steps, the pH at which the reaction steps are performed, and the methods used to purify the IgG-HRP conjugate and remove the unconjugated enzyme [4,5,16,19].

In this work, individual reaction steps during the preparation of the IgG-HRP conjugate were monitored by analyzing the starting samples, intermediates, and final product with multi-detection SEC (Figure 2). The absolute molecular weight of IgG and HRP as input compounds was in accordance with the expected values (Table 1). Their purity was exceptionally high (>92%) and more than suitable for the coupling reaction.

The results showed that the oxidation of the polysaccharide residues of HRP with sodium periodate did not affect the *Mw* or composition of the sample peaks. In addition, the reaction did not induce any self-coupling of the activated HRP (Appendix A and Appendix A). Although HRP contains only two available primary *ε*-amine groups (two lysine residues) and its self-coupling is not a common phenomenon [1], the risk was minimized by keeping the activated enzyme in a slightly acidic medium [16].

A multi-detection SEC analysis (Figure 1 and Figure 2) showed that the conjugation reaction produced a heterogeneous sample containing IgG-HRP conjugates with the average stoichiometry of 1:2, but also unwanted populations, composed either of aggregates or higher *Mw* conjugation products. The possible reason for their occurrence might be the formation of a large number of highly active aldehyde groups on the HRP molecules [6,18] which, when activated, bridge the IgGs, causing their cross-linking [19,20]. Nakane and Kawaoi demonstrated that one IgG molecule can bind a maximum of five to six activated HRP molecules [6]. The excessive cross-linking could be decreased by reducing the concentration of IgG and/or HRP, adjusting the IgG/HRP molar ratio, or shortening the reaction time. In order to optimize the reaction conditions and improve the yield, the influence of these factors on the conjugation could be easily monitored by multi-detection SEC, which, in only one analysis, generates multiple quantitative data of the sample, including absolute *Mw*, polydispersity, and composition.

In addition, the *Mw* or composition of the sample peaks were not affected by the reduction step. However, the ultrafiltration step led to the increase of aggregate content (Table 2, Figure 2), possibly as a consequence of a combination of several factors, such as the high protein concentration, enhanced solution viscosity, and physical stress, which implies the filtration rate, fluid flow-induced shear stress, interfacial interactions, and extensive contact with the membrane surface [21,22,23,24]. Such a finding suggests that ultrafiltration might not be the best method for the matrix exchange and goes in favor of dialysis, as described in the majority of published protocols [4,6,16,17]. Namely, the ultrafiltration process involves a transient increase in the concentration of the sample in contrast to dialysis, which might be a trigger for aggregation.

In addition to the characterization of our in-house anti-Vero IgG-HRP conjugate, another aim was to gain insight into its compliance with those that are commercially available. For that purpose, we have randomly selected three conjugates from two manufacturers. The multi-detection SEC chromatograms of the purchased conjugates showed very similar profiles which indicates consistency in their preparation (Figure 4, Table 3). All three conjugates contained a low share of two types of IgG-HRP conjugates (<19%) and a significant amount of free IgGs (42–62%). In contrast, our in-house preparation had a greater share of the IgG-HRP conjugates (32%) than the purchased ones and fully coupled IgGs. Additionally, the aggregates’ share appeared to be roughly three-fold lower in the purchased conjugates than in ours. The observed differences are probably the consequence of variations in the protocols that were used for their preparation. However, the applicability of multi-detection SEC, rather than a comparison of different conjugation reaction processes, was the main focus of this paper. It is interesting to note that the estimated *Mw* of the putative IgG monomer in two commercial conjugates, both prepared from rabbit IgGs, was ~144 kDa which is in complete agreement with the *Mw* of the rabbit IgG previously determined by Rayner et al. [25].

The optimal relative size of the antibody–enzyme complex preferably should be adapted to the assay application. Nakane and Kawaoi demonstrated that conjugates containing an average of two activated HRP molecules per one IgG molecule were more suitable than highly labelled conjugates (with more than four activated HRP molecules per one IgG molecule) for immunohistochemical staining where penetration of the complex through membrane barriers is an important consideration [6]. In contrast, high-molecular weight IgG-HRP conjugates could be an appropriate choice for ELISA procedures, where high sensitivity is important, and washing off excess conjugate is not a problem. However, two critical points still remain. Firstly, the aggregates reduce the active ingredient share [5]. Secondly, the presence of free IgGs affects the sensitivity of immunoassays. Therefore, the composition of the IgG-HRP conjugate sample directly influences the degree of its quality as a reagent.

The functionality of the prepared in-house anti-Vero IgG-HRP conjugate, as well as its optimal working dilution for ELISA, was determined. It is a well-known fact that the optimal dilutions of commercially available IgG-HRP conjugates vary, irrespective of the declared concentration. According to our results, the conjugates’ working range might be influenced by the differences in their content as well, primarily the share of the (enzymatic/immunological) active substance and aggregates, affecting the sensitivity and specificity of the final product.

In this work, it was demonstrated that advanced molecular characterization with the multi-detection SEC could offer an in-depth insight into the conjugation reaction. Many other standard manufacturing procedures could also benefit from its implementation. Multi-detection SEC might serve as a routine technique in quality control or provide guidelines that could contribute to the process improvement, providing full molecular characterization due to its ability to simultaneously measure and characterize the sample’s composition on multiple levels.

## 4. Materials and Methods

### 4.1. Reagents and Chemicals

Horseradish peroxidase and NaBH_4_ were from Sigma-Aldrich (St. Louis, MO, USA). NaIO_4_ was from Merck (Darmstadt, Germany). A daily supply of ultrapure water was obtained from a PureLab Classic purification system (Elga, High Wycombe, UK). Chemicals for buffers and solutions were from Kemika (Zagreb, Croatia), unless stated otherwise.

Three different commercially available IgG-HRP conjugates from two manufacturers were analyzed with multi-detection SEC: anti-goat IgG-HRP produced in rabbit (IgG-HRP 1), anti-rabbit IgG-HRP produced in goat (IgG-HRP 2), and anti-guinea pig IgG-HRP produced in rabbit (IgG-HRP 3).

### 4.2. Purification of Antibodies

Antibodies were purified from a guinea pig (immunized with lysate of Vero cells) serum using protein A HiTrap MabSelect Xtra column (1 mL; GE Healthcare, Chicago, IL, USA). Serum was 2-fold diluted with the binding buffer (20 mM phosphate buffer, 0.15 M NaCl, pH 7.2) and loaded (2 mL/run) to the pre-equilibrated protein A column at a flow rate of 2 mL/min on an ÄKTApurifier 100 system equipped with P-900, UV-900, and pH/C-900 (GE Healthcare, USA) at room temperature (RT). The absorbance was monitored at 280 nm. The bound antibodies were eluted from the column with 20 mM glycine, 0.15 M NaCl, and pH 2.3. Eluted fractions from all runs were pooled, concentrated, and transferred to the binding buffer using a Vivacell device (Sartorius, Göttingen, Germany) with a 50 kDa molecular weight cut-off (MWCO) polyethersulfone membrane, resulting in polyclonal anti-Vero IgG sample that was used as a reagent for preparation of the IgG-HRP conjugate and for plate coating in the ELISA.

### 4.3. Protein Concentration Estimation

All protein (antibodies and total Vero cell lysate proteins) concentrations were estimated spectrophotometrically using the following equation:

*γ* [mg/mL] = (A_228.5 nm_ − A_234.5 nm_) × *f* × dilution, where Ehresmann’s factor “*f*” of 0.3175 was used [26]. Appropriate dilution of each sample was independently prepared a minimum of three times to obtain the mean value for further calculation of yield and purity. Absorbance measurements were performed on a Multiskan Spectrum (Thermo Fisher Scientific, Waltham, MA, USA).

### 4.4. Preparation of IgG-HRP Conjugate

Conjugate of HRP and guinea pig polyclonal antibodies was prepared by a modified periodate oxidation method [3,4,16,17]. Briefly, freshly prepared 0.1 M sodium (meta) periodate solution (220 equivalents) was added to 4 mg/mL HRP solution (1 equivalent) in water and stirred gently on a shaker (300 rpm) for 20 min at room temperature (RT) in dark. Excess of periodate was removed by ultrafiltration of the activated HRP solution against 1 mM sodium acetate buffer, pH 4.4, in centrifugal filter unit with 10 kDa MWCO membrane (Sartorius, Göttingen, Germany). Before conjugation reaction, pH of ultrafiltrated and activated HPR solution was adjusted to alkaline pH by adding 0.2 M sodium carbonate buffer, pH 9.5 (40 μL to 1 mL HRP solution). Moreover, anti-Vero IgG sample was transferred into 10 mM sodium carbonate buffer, pH 9.5, using a 50 kDa MWCO ultrafiltration device (Sartorius, Göttingen, Germany). Then, 4 mg/mL IgG solution (20 mg, 1 equivalent) was added to the activated HPR solution (2 equivalents) and stirred on a shaker (300 rpm) for 2 h at RT in dark. After the addition of 4 mg/mL freshly prepared sodium borohydride solution (210 equivalents), the mixture was stirred (300 rpm) for 90 min at RT in dark. Following filtration through a 0.45 μm PVDF syringe filter (Merck Millipore, Darmstadt, Germany), solution of IgG-HRP conjugate was ultrafiltrated and transferred to PBS using a 100 kDa MWCO ultrafiltration device (Sartorius, Göttingen, Germany). Aliquots of starting samples, all intermediates, and final product were collected for multi-detection SEC and protein concentration estimation.

### 4.5. Multi-Detection SEC

Size exclusion chromatography was carried out on TSKGel G3000SWXL column (7.8 × 300 mm; Tosoh Bioscience, Japan) with a TSK gel SWXL guard column (6.0 × 40 mm; Tosoh Bioscience, Japan). HPLC analysis was performed with 0.1 M phosphate–sulphate running buffer, pH 6.6, at a flow rate of 1 mL/min at RT on a HPLC Prominence system (Shimadzu, Kyoto, Japan). The samples were centrifuged (Eppendorf, Hamburg, Germany) at 3000 × *g* for 10 min and loaded to the column in a volume of 50 µL/run. Total run time was 17 min. Advanced detection was obtained on Omnisec Reveal (Malvern Panalytical Ltd., Malvern, United Kingdom) multi-detector module which consists of refractive index (RI) detector, UV/Vis absorbance detector, hybrid right-angle light scattering, low-angle light scattering (RALS/LALS) detector, and intrinsic viscosity detector. Calibration was performed using a bovine serum albumin (BSA) standard (Thermo Scientific, Waltham, MA, USA). The resulting chromatograms were analyzed using RI, RALS, LALS, and viscometer detectors. The d*n*/d*c* values, calculated from the sample concentrations, were used to determine the molecular weight of the peaks using Omnisec software (version 11.32). Each sample was analyzed at least twice.

### 4.6. ELISA for Evaluation of IgG-HRP Conjugate Functionality

Indirect ELISA was performed according to the previously published procedure [27]. Briefly, flat-bottomed high-binding ELISA plate (Corning, New York, SAD) was coated with 100 μL/well of guinea pig anti-Vero IgG sample (1 μg/mL or 10 μg/mL) in 0.05 M carbonate buffer, pH 9.0, and incubated overnight at RT. The plate was washed and blocked with 250 μL/well of 2% (*w/v*) BSA (Sigma-Aldrich, Saint Louis, MO, USA) in PBS with 0.05% (*v*/*v*) Tween 20 for 2 h at 37 °C. After blocking, a lysate of Vero cells of known concentration was applied as a standard in two-fold serial dilutions (100 μL/well) in duplicates, starting from 30 ng/μL and incubated overnight at RT. Plates were washed and 100 μL/well of guinea pig anti-Vero IgG-HRP conjugate in 1:1000, 1:2000, 1:4000, 1:16,000, or 1:32,000 dilution was added and incubated for 2 h at 37 °C. Following washing, 100 μL/well of 0.6 mg/mL *o*-phenylenediamine dihydrochloride (OPD) solution in citrate–phosphate buffer, pH 5.0, containing 0.5 μL 30% H_2_O_2_/mL of OPD solution, was added and incubated for 30 min at RT in the dark. The enzymatic reaction was stopped by addition of 50 μL/well of 12.5% (*w*/*v*) H_2_SO_4_ and the absorbance was measured on a Multiskan Spectrum (Thermo Fisher Scientific, Waltham, MA, USA) at 492 nm.

## 5. Conclusions

In this study, we investigated the potential of multi-detection SEC as a tool for monitoring the conjugation reaction between the antibody and HRP, as well as the advanced characterization of prepared or commercially available IgG-HRP conjugates. The absolute molecular weight, polydispersity, size, composition of sample peaks, and aggregate content were determined in all reaction stages. Based on the absolute molecular weight, the average stoichiometric relationship between the IgG and HRP in the conjugate was estimated. This study showed that multi-detection SEC could be an appropriate method for monitoring the conjugation reaction in order to gain insight into the process quality with the aim of the enhancement of the active component’s yield, and, consequently, the improvement of the reagent’s sensitivity and specificity. Furthermore, it seems to be a promising and powerful technology for the quality control of many other protein–protein conjugates, providing a detailed, precise, and full molecular characterization.

## Figures and Tables

**Figure 1 molecules-28-04567-f001:**
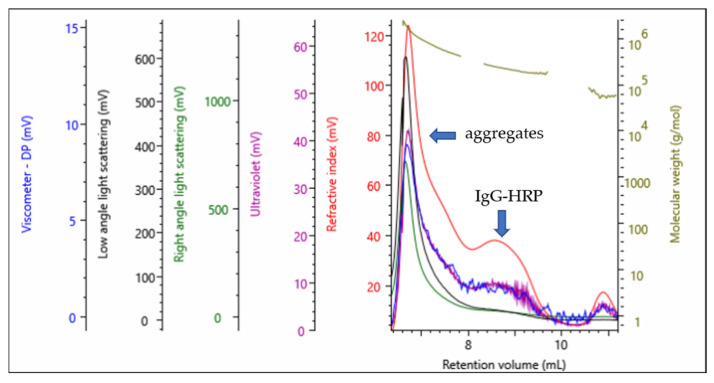
Representative multi-detection SEC chromatogram of the UF IgG-HRP sample—refractive index (red), right-angle light scattering (green), low-angle light scattering (black), viscometer (blue), and ultraviolet (purple). The molecular weight of each species is shown in olive.

**Figure 2 molecules-28-04567-f002:**
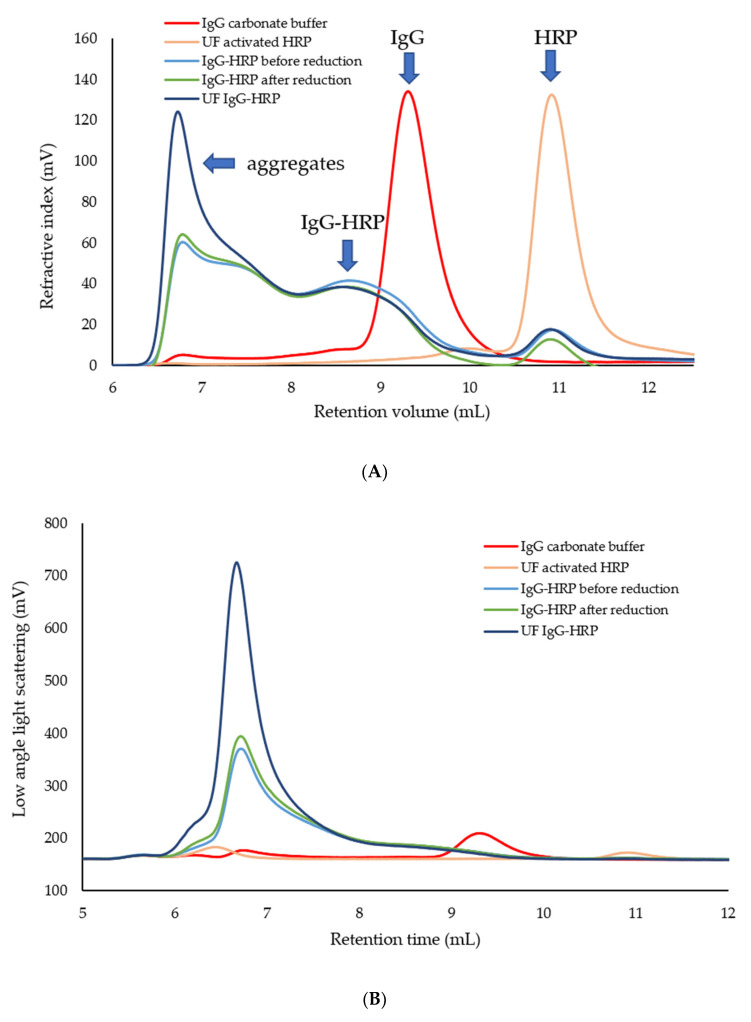
Overlay of multi-detection SEC chromatograms of starting samples (IgG and HRP), intermediates, and final IgG-HRP product (~1.5 mg/mL) obtained by (**A**) refractive index (RI) or (**B**) low-angle light scattering (LALS) detector.

**Figure 3 molecules-28-04567-f003:**
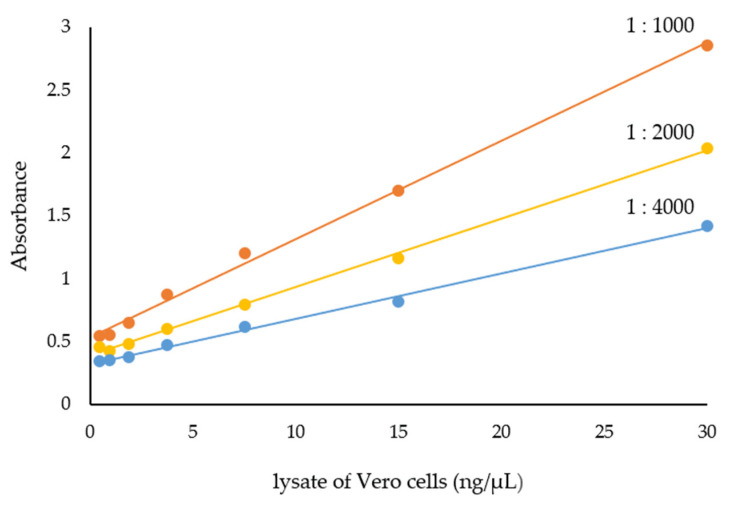
Standard curves for several dilutions of anti-Vero IgG-HRP conjugate in the ELISA for quantification of total Vero cell lysate proteins. The plate was coated with 10 μg/mL of anti-Vero IgG.

**Figure 4 molecules-28-04567-f004:**
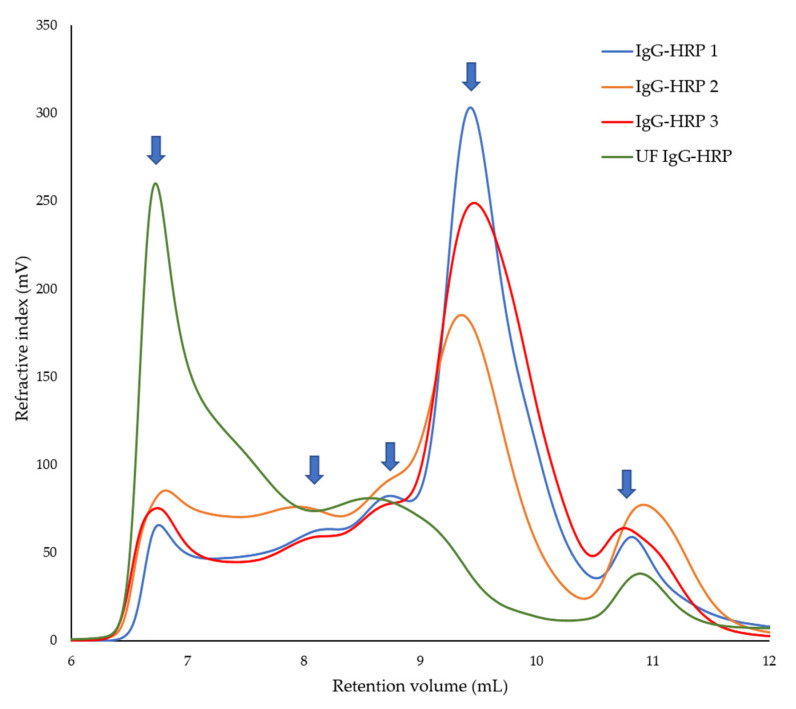
Overlay of RI chromatograms of commercially available IgG-HRP conjugates and prepared UF IgG-HRP conjugate (~3 mg/mL).

**Table 1 molecules-28-04567-t001:** Molecular characterization data for IgG monomer, HRP monomer, and final product (IgG-HRP conjugate) obtained by multi-detection SEC. The results are expressed as mean value ± standard error (SE) from *n* measurements.

Sample	*n*	*V_R_* (mL)	*Mw* (kDa)	*Mw*/*Mn*	*IV* (dL/g)	*Rh* (nm)	Share (%)
IgG	3	9.3	152.5 ± 1.2	1.003 ± 0.000	0.042 ± 0.002	4.67 ± 0.09	92.0 ± 0.2
HRP	4	10.9	43.1 ± 0.5	1.004 ± 0.000	0.032 ± 0.001	2.79 ± 0.01	93.8 ± 0.7
IgG-HRP	5	8.6	234.9 ± 3.2	1.028 ± 0.005	0.047 ± 0.002	5.55 ± 0.09	32.1 ± 2.2

**Table 2 molecules-28-04567-t002:** Molecular characterization data for the IgG-HRP samples before/after reduction and the final product (UF IgG-HRP) obtained by multi-detection SEC. The results are expressed as mean value ± standard error (SE) from *n* measurements.

	Identity	*V_R_* (mL)	*Mw* (kDa)	*Mw*/*Mn*	Share (%)
IgG-HRP	aggregate	6.8	902.2 ± 34.0	1.200 ± 0.016	52.76 ± 1.34
before reduction	IgG-HRP	8.6	232.2 ± 5.1	1.021 ± 0.002	39.59 ± 1.62
(*n* = 2)	n/d *	10.9	71.8 ± 0.9	1.019 ± 0.004	7.66 ± 0.28
IgG-HRP	aggregate	6.8	983.1 ± 51.5	1.239 ± 0.050	55.47 ± 1.46
after reduction	IgG-HRP	8.6	246.4 ± 0.9	1.022 ± 0.003	38.64 ± 1.88
(*n* = 2)	n/d	10.9	83.3 ± 5.6	1.011 ± 0.004	5.90 ± 0.42
UF IgG-HRP	aggregate	6.8	1273.4 ± 49.4	1.285 ± 0.045	61.82 ± 2.52
(*n* = 5)	IgG-HRP	8.6	234.9 ± 3.2	1.028 ± 0.050	32.06 ± 2.25
	n/d	10.9	64.0 ± 0.6	1.015 ± 0.002	6.12 ± 0.41

* n/d—not defined.

**Table 3 molecules-28-04567-t003:** Molecular characterization data for commercially available IgG-HRP conjugates from multi-detection SEC analysis. The results are expressed as mean value ± standard error (SE) from 3 measurements.

	*V_R_* (mL)	*Mw* (kDa)	*Mw*/*Mn*	*IV* (dL/g)	*Rh* (nm)	Share (%)
IgG-HRP 1 (rabbit anti-goat IgG-HRP)	7.1	965.0 ± 9.2	1.279 ± 0.016	0.076 ± 0.007	10.20 ± 0.35	16.18 ± 1.01
	8.2	321.2 ± 0.7	1.003 ± 0.001	0.059 ± 0.005	6.69 ± 0.21	5.20 ± 0.54
	8.7	225.2 ± 0.6	1.005 ± 0.001	0.054 ± 0.005	5.75 ± 0.18	9.13 ± 1.21
	9.4	145.5 ± 0.4	1.007 ± 0.001	0.046 ± 0.003	4.74 ± 0.10	61.94 ± 0.89
	10.8	68.5 ± 2.1	1.009 ± 0.004	0.040 ± 0.006	3.50 ± 0.20	7.52 ± 0.95
IgG-HRP 2 (goat anti-rabbit IgG-HRP)	6.8	1175.7 ± 5.6	1.205 ± 0.019	0.080 ± 0.000	11.20 ± 0.02	21.27 ± 0.47
	8.0	371.4 ± 0.9	1.006 ± 0.001	0.061 ± 0.001	7.09 ± 0.03	9.71 ± 0.22
	8.7	236.1 ± 0.6	1.005 ± 0.000	0.053 ± 0.000	5.82 ± 0.02	8.99 ± 0.38
	9.4	156.6 ± 0.4	1.005 ± 0.001	0.046 ± 0.001	4.85 ± 0.03	42.36 ± 0.33
	10.9	55.6 ± 0.6	1.032 ± 0.004	0.034 ± 0.002	3.11 ± 0.04	17.67 ± 0.19
IgG-HRP 3 (rabbit anti-guinea pig IgG-HRP)	6.7	1754.5 ± 7.0	1.348 ± 0.045	0.078 ± 0.002	12.50 ± 0.16	15.75 ± 0.99
	8.2	353.2 ± 0.4	1.013 ± 0.003	0.064 ± 0.003	7.08 ± 0.11	8.58 ± 0.73
	8.7	223.7 ± 0.5	1.003 ± 0.001	0.055 ± 0.002	5.78 ± 0.09	6.97 ± 0.67
	9.4	144.8 ± 0.1	1.008 ± 0.001	0.046 ± 0.001	4.74 ± 0.03	56.16 ± 0.77
	10.8	62.7 ± 0.6	1.017 ± 0.004	0.041 ± 0.002	3.43 ± 0.07	12.55 ± 0.57

## Data Availability

The data presented in this study are available upon request from the corresponding author.

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
