# Peer review of "Multi-Detection Size Exclusion Chromatography as an Advanced Tool for Monitoring Enzyme–Antibody Conjugation Reaction and Quality Control of a Final Product"

_molecules, 2023, doi:10.3390/molecules28114567_

Round 1

Reviewer 1 Report

The paper investigates the potential and suitability of multi-detection size exclusion chromatography (SEC) for monitoring enzyme-antibody conjugation reaction and quality control of the final product. The authors prepared guinea pig anti-Vero IgG-HRP conjugate using a modified periodate oxidation method, and analyzed the molecular characterization data of starting samples, intermediates, and final products by multi-detection SEC. The study demonstrates that this methodology is a promising and powerful technology for IgG-HRP conjugate process control and development, and for the quality control of the final product.

In general, I found the paper shows a thorough investigation of the potential and suitability of multi-detection SEC for monitoring enzyme-antibody conjugation reaction and quality control of the final product and provides comprehensive insight into the molecular characterization data of the protein sample, including absolute molecular weight, molecular weight distribution or polydispersity, molecular size, intrinsic viscosity, molecular structure, and recovery. Otherwise, the paper demonstrates the plausibility of the application of multi-detection SEC in the quality control of the final product by analysis of several commercially available reagents. However, this manuscript has some of the trivial discrepancies which will be described below. I feel, once they are rectified, the paper is suitable for acceptance and publishing.

1. The article could benefit from including a more detailed description of the intermediate structures and providing a diagram illustrating the chemical reaction.

2. The study lacks proper evaluation of the proposed method, as no comparisons or discussions with widely-known baselines in the field are provided. how confident are the authors that the proposed method is superior to existing methods in monitoring enzyme-antibody conjugation reaction and quality control of the final product?

The paper investigates the potential and suitability of multi-detection size exclusion chromatography (SEC) for monitoring enzyme-antibody conjugation reaction and quality control of the final product. The authors prepared guinea pig anti-Vero IgG-HRP conjugate using a modified periodate oxidation method, and analyzed the molecular characterization data of starting samples, intermediates, and final products by multi-detection SEC. The study demonstrates that this methodology is a promising and powerful technology for IgG-HRP conjugate process control and development, and for the quality control of the final product.

In general, I found the paper shows a thorough investigation of the potential and suitability of multi-detection SEC for monitoring enzyme-antibody conjugation reaction and quality control of the final product and provides comprehensive insight into the molecular characterization data of the protein sample, including absolute molecular weight, molecular weight distribution or polydispersity, molecular size, intrinsic viscosity, molecular structure, and recovery. Otherwise, the paper demonstrates the plausibility of the application of multi-detection SEC in the quality control of the final product by analysis of several commercially available reagents. However, this manuscript has some of the trivial discrepancies which will be described below. I feel, once they are rectified, the paper is suitable for acceptance and publishing.

1. The article could benefit from including a more detailed description of the intermediate structures and providing a diagram illustrating the chemical reaction.

2. The study lacks proper evaluation of the proposed method, as no comparisons or discussions with widely-known baselines in the field are provided. how confident are the authors that the proposed method is superior to existing methods in monitoring enzyme-antibody conjugation reaction and quality control of the final product?

Reviewer 2 Report

Quality control of enzyme conjugates is an inevitable part of manufacturing of ELISAs. Herein, the Authors present their results on application of multi-detector SEC for characterization of intermediates and obtained conjugates. Besides, three commercial conjugates were also compared by SEC. The article is well written and clear. I fing it interesting for those who develop ELISA tests and seek for options of internal quality control. However, in my opinion the advantades of the most of the detecting methods are not shown properly.

The authors measured several parameters: UV, refractive index, internal viscosity, light scattering (2 detectors), but it is not clear what valualble information can give most of them in terms of conjugate properties. For example let's see the table 1. In line with molecular weigth of each fraction, IV (internal viscosity?), Rh (hydrodinamic radius?) are also reported. But what should reader do with these data? The Authors do not comment on them. What kind of information about properties of molecules/aggregates can be drawn from these data? The main feature of multi-detector SEC is utilization of several special detectors, other than commonly used UV. But the authors hardly take advantage any of them. For example, after reading this paper I did not understand what is difference between LALS and RALS and what specific knowledge about structure of conjugates they produce. What properties of conjugates/IGGs/enzyme can be characterized by refractive index detector? In total, my impression is that title does not reflect the contents of the paper. The same results can be obtained using SEC with UV detector (share of fractions with different molecular weight). Advantages of different detectors are not demonstrated. My opinion is that paper should be changed and resubmitted.

Comments:

1. Line 353 - Vero cell lysate concentration. Is it concentration of total protein? How did you determine it?

2. Table 1. Include abbreviations (IV, Rh and so on)

3.Line 342 - What is dn/dc?

Reviewer 3 Report

The manuscript describes the analytical and functional characterization of a conjugate of horseradish peroxidase and guinea pig anti-Vero IgG. Analyses were performed by size exclusion chromatography with multiple detectors, whereas functional characterization of the product was evaluated by ELISA.

Data are overall sound, but I recommend the publication of the manuscript after a revision of English grammar since several parts of the draft are poorly understandable and require rewording.

The major issues that have to be fixed before publication are:

1- Variability (e.g., standard deviation) is missing from data reported in supporting information (table S1, S2 and S3) and in some section of the text, like from line 69 to 73 (and in some following sections). Moreover, most of the author’s conclusions (e.g., line 84 “Oxidation of polysaccharide residues of the HRP didn’t affect its Mw”) are based on numerical considerations since statistic tests that support author’s conclusions are missing.

2—the sentence from line 32 to line 34 is puzzling. The authors stated that HRP is a glycoprotein with a “broad specificity that allows to be measured by absorption, fluorescence, and luminescence”. This sentence is puzzling for several reasons. First, what kind of specificity are the author talking about? Second, if the so-called “specificity” allows measurements with so many different techniques it sounds more like there is no specificity at all. A rewording of the sentence would be nice to avoid misunderstandings.

3—What is Mn? It is mentioned for the first time at line 48 to define the parameter Mw/Mn, and many other times later on in the manuscript. However, there is no definition of what Mn is in the text. A list of abbreviations would improve manuscript reading.

4— statements from line 51 to 54 are not correct. RI and UV detectors do not measure the sample’s concentration and composition, but instead parameters from which sample’s concentration and composition can be deducted. In fact, when describing the analytical procedures, the authors indicated “Antibody concentration estimation” as title of section 4.3. Similarly, SLS detector does not reveals absolute molecular weight but can be used to measure an apparent molecular weight since protein light scattering depends on molecular weight but also conformation, pH, ionic strength and so on (see table S1 where the calculated molecular weight is changing when measured in phosphate or carbonate buffer). The same concepts are repeated multiple times in the text (e.g., in the conclusion section). I encourage the authors to be more cautious and precise about the definition of the measurements done throughout the text.

I recommend a revision of English grammar since several parts of the draft are poorly understandable and require rewording.

Round 2

Reviewer 2 Report

The Authors added excellent description of multi-detector SEC functionality. Now, the paper can be published with minor corrections.

Comment

It would be much clearer to the reader if the Authors signed corresponding fraction for each arrow in Fig. 4

Typos:

Line 142 - borohy-dride

Line 140-143 - text formatting

Line 166 -  <1.2 or >1.2?